# Validity of Dynamic Capabilities in the Operation Based on New Sustainability Narratives on Nature Tourism SMEs and Clusters

**Alejandro J. Gutiérrez Rodríguez [1,2,]\*, Nini Johanna Barón [1] and José Manuel Guaita Martínez [3]**

1    Faculty of Economics and Administrative Sciences, Ibague University, 73002 Ibague, Colombia; johanna.baron@unibague.edu.co
2    School of management, Rosario University, 111711 Bogota, Colombia
3    Faculty of Business, Valencian International University, Career del Pintor Sorolla, 21, 46002 Valencia, Spain; josemanuel.guaita@campusviu.es
*    Correspondence: alejandro.gutierrez@unibague.edu.co; Tel.: +57-310-210-8234

**Abstract:** This study aims to validate the relationships between the dynamic capabilities in the operation of small and medium enterprises (SMEs) that constitute the Nature Tourism Cluster (also known as "ecotourism") in Colombia, through the application of surveys to managers and owners of hotels and lodgings of rural tourism, travel agencies, tour guides and operators, bars, restaurants and tourist transport centers, whose data obtained, support our hypothesis that the dynamic capacities of absorption, adaptation and innovation influence the functioning of SMEs, while at the cluster level, there is an positive relationship in the interaction of absorption and innovation capabilities. The greatest contribution of our research consists in the development of an empirical study that is based on the main contributions of the dynamic capabilities promoted by Teece (absorption, adaptation and innovation), and that allowed to determine the degree of influence that managers have to take decisions and undertake sustainable ecotourism actions, both at the SME level and at the cluster level. For this reason, our research provides a better understanding of how dynamic capabilities operate at the individual commercial level, as well as at the cluster level, in the combination of absorption, adaptation and innovation capabilities to foster new sustainability narratives and maintain sustainable ecotourism. Our results also point out the limitations and challenges for the sustainable tourism sector in Colombia.

**Keywords:** innovation; adaptation; absorption; small and medium enterprises; cluster; new sustainability narratives; competitive advantage

## 1. Introduction

The increase in business activities that involve collaborations for the purpose of penetrating sophisticated tourism markets, both local and international, has generated substantial interest in the academic community in recent years [1,2]. To succeed in this type of business endeavor, companies need to develop capabilities that build sustainable competitive advantages, allow them to adapt to changing market conditions, enhance learning from experiences, develop practices to identify opportunities and trends in the relevant business environments, and pilot and develop new goods and services that are based on new technologies [3–7].

A country's ability to compete for investment in tourism may be measured by the World Economic Forum's Travel & Tourism Competitiveness Index (TTCI), which has recently been updated with a new methodology [8]. Another relevant variable in this type of analysis is the degree of seasonality of

a group of tourist destinations [9–12]. In both cases, the global mega-trend highlights the importance of developing competitive advantages that take advantage of the local wealth of its natural resources, without affecting communities and biodiversity, so that sustainable tourism is carried out, which mitigates the high tourist densities and that promotes sustainable practices in the rational use of resources such as water and energy, as well as actions aimed at reducing $CO_2$ emissions [13–15].

From the nature tourism trend, the concept of ecotourism arises, composed of three dimensions, based on nature, environmental education and management in a sustainable manner, and governed by two principles of sustainability (I) to support economies local and (II) to support conservation [16]. Thus, ecotourism is a subcomponent in the domain of sustainable tourism and aims to achieve sustainable development by planning and managing successfully to achieve its social and environmental objectives [17].

For this reason, organizations (profit and non-profit), should generate an awareness on the part of the owners and managers, in decision-making and in the actions to be undertaken around sustainable tourism, more specifically in ecotourism, and that demand to constitute in the firms, internal and external capacities, which we highlight in this investigation, within the framework of the dynamic capabilities [18–20].

The theory of the dynamic capacities of companies has undergone significant theoretical and empirical advances, focusing on ways in which qualities such as innovation, adaptation, and absorption influence companies' operations and performance. Recently, the role of dynamic capabilities in terms of collaborative systems has attracted the attention of researchers, particularly in theoretical and empirical analyses conducted on the influence of dynamic capabilities in any type of organization (network, company, or group) in obtaining superior returns that are backed by a sustainable competitive advantage [21–25].

This study empirically analyzes the dynamic capabilities of absorption, adaptation and innovation, such as the development of internal and external skills in the processes of organizations, in order to integrate and adapt to the rapid changes of highly competitive business environments [16,17,19,20]. It joins empirical studies with a new approach to business management and related to the business and sustainable growth of business management in the tourism sector in the Tolima region (Colombia), and its new challenges after being a region of armed conflict for 60 years, it aroused the need to establish new sustainability narratives. As analyzed factors, this document investigates whether these dynamic capabilities are significant, not only to create competitive advantages with relevant impacts in the areas of strategic management and business management, but also to measure the influence of new sustainability narratives from the dynamic capabilities perspective, in decision-making focused on strengthening local ecotourism, both in SMEs and at the cluster level, which allows a balance between taking strategic actions to care for the environment and the economic and social development of the communities [26,27].

Therefore, this study aims to empirically validate the relationship of the dynamic capabilities of companies included in the nature tourism cluster of the Department of Tolima (Colombia), focusing on complex schemes, such as clusters, and using instruments and measurement scales that we allow to infer the scope of the relationship between each type of dynamic capacity.

## 2. Literature Review

### 2.1. Dynamic Capabilities

Dynamic capabilities are the competencies that an organization possesses that allow it to undertake a constructive change in response to the uncertainty of changing markets [24,26] by facilitating the adaptation, integration, and configuration of internal and external resources. Although dynamic capabilities lack a unique definition, they are characterized by being tacit, causally ambiguous, organization-specific, socially complex, and route-dependent [28,29]. Since they are difficult to

distinguish and imitate, they function as a protective barrier and contribute to an organization's competitive advantages [8,19,24,30–32].

Some primary examples of dynamic capabilities include detection, capture, and transformation capabilities [33]; product development; and strategic decision-making and alliance-building [34]. However, the most cited and studied globally are the generic absorption, adaptation, and innovation capabilities proposed by Teece and Pisano (1994), and Teece, Pisano, and Shuen (1997).

With the support of recent studies by David Teece (2016, 2018), which support the relevance of dynamic capabilities in changes in the business environment, the Karadag analysis (2019), of the evolution of dynamic capabilities from the theoretical beginning (Teece, et al. Al., 1994, 1997), to this day, show the need to develop more empirical studies that contribute to theoretical application, at the level of companies and other organizations with current and complex environments, due to its business and sector dynamics [35].

This means, on the one hand conducting an analysis of the factors that influence the most dynamic sectors with the greatest potential for sustainability in countries sustainable economic growth and development. On the other hand, include analysis of specific sectors, such as tourism, and in different countries.

## 2.2. Relationships between Dynamic Capabilities for SMEs and Clusters

In recent studies, dynamic capabilities have been analyzed in terms of small and medium enterprises' (SMEs) adaptability and innovation with respect to their products/services (PES). Findings indicate the need to develop knowledge networks that allow SMEs to analyze and understand the dynamics of an industry, and to support the SMEs' planning and long-term vision so that they can develop processes for strategic adaptation and innovation with respect to their products or services, based on the needs of their customers [36].

Similarly, the dynamic capacities that organizations obtain through networks and knowledge groups that allow them to achieve long-term structural changes through adaptation and innovation have been highlighted. For example, Crespo, Suire, and Vicente (2016), combine evolutionary economic geography and a network-based clustering approach to identify the contribution and influence of hierarchy (distribution of degrees) and substitution (correlation of degrees) on the innovative capabilities of groups throughout the life cycle of the industry, focusing on the mobile phone industry in Europe from 1988 to 2008 [37].

In a similar case, studies such as that of the ceramics industry in Brazil are observed, highlighting the dynamic capacities at different levels, such as the adaptation strategies that actors take at the micro dynamic level (companies), based learning processes in networks at the dynamic meso level (sector) and evolution by invoking institutions at the dynamic macro level (region) [38], capabilities that are also observed in other studies that analyze the impact on global business (GB) and entrepreneurship international (EI). In both cases, it can be ensured that adaptation, absorption and innovation capabilities co-create a business model in organizations, which is recognized by some authors, such as the vision of dynamic capabilities (VDC), In both cases, it can be ensured that adaptation, absorption and innovation capabilities co-create a business model in organizations, which is recognized by some authors, such as the vision of dynamic capabilities (VDC), model that It allows managing and shaping networks effectively, particularly in the face of turbulence in emerging markets [39]. Dynamic capacities have had a geographical evolution, even in different regions, different authors have studied the relevance of the capacities to innovate, of the learning dynamics of individuals and companies, and of adaptation within a global network, especially in cases such as the role of multinationals in a region, the entry and exit of companies in a sector, as well as social capital, which have influenced factors relevant to sustainable development [36,39,40].

Various theoretical perspectives highlight common aspects of dynamic capabilities, such as:

- Innovation: dynamic capacity based on a company's ability to innovate in marketing new products, improve goods and services and key processes of the company, and generate new business opportunities [24,26,29,30,40].
- The evolutionary: an alteration and/or recombination of competencies, resources and processes that results from absorbing new knowledge, information, resources or assets, which in turn, are capable of generating new organizational skills, facilitating learning in the company [26,27,30,41].
- The contingent: the capacity to react and make strategic adjustments in response to changes in the environment; that is the company's ability to adapt to new environmental conditions [26,27,30,31,41].

In sum, it is logical to think that academic literature supports the idea that dynamic capabilities drive economic development in the increasingly variant of business environments. This new role of companies translates into a transformation of new business organizations into adaptive institutions, creators of new businesses and with constant self-learning procedures.

### 2.3. Relationship between Absorption and Adaptation as Dynamic Capabilities

Cohen and Levinthal (1989) define absorption capacity as an internal mechanism that allows a company to analyze various inputs, both external and internal, and, depending on its current technical knowledge, to identify the most appropriate information to adopt, depending on the R&D and technological development and needs of the company. This requires a process of repetitive experimentation, which facilitates better execution, better performance and faster technical learning [7,41–44]. Recently, authors such as Essid and Berland (2018), highlight the importance of dynamic absorption capacity in French companies, which have allowed the people of the company to manage knowledge to take measures in the operation of companies, aimed to process optimization, resource reconfiguration and development of new products, which mitigate the impact with the environment [26].

Adaptive capacity is responsible for identifying and taking advantage of opportunities presented in the market [30,45]. It allows firms to engage in activities such as combining, reconfiguring and integrating resources and processes, promoting the systemic development of other capacities, since it allows a company to evolve its organizational form, give flexibility to its structures in the use of strategic resources, and in segmenting the needs of a changing market [3,31,44,45]. The dynamic capacity for adaptation has been studied in the last decade in ecotourism companies, which demonstrates that environmental benefits and economic returns can be achieved by linking the expectations of stakeholders [20] in corporate governance processes [17,19].

When identifying absorption and adaptation capacity in enterprises, special characteristics of each sector and country should be taken into account in investigating those factors that promote sustainable development. One of the future steps in this line of research would be to make a comparison whiting the tourism sector in different countries.

### 2.4. Relationship between Adaptation and Innovation Dynamic Capabilities

The capacity for innovation focuses on the ability to develop new products and/or markets, through adjustments in a company's strategic orientation using innovative behaviors and processes. It is complemented by other dynamic capabilities in mobilizing resources to collect existing knowledge within the company, and to obtain new knowledge that is translated into innovations of products, processes, strategies, marketing and management to achieve a continuous interaction between the business and market demands [23,30,44–47].

The capacity for innovation requires the optimal adjustment of processes to accommodate the timely reconfiguration and combination of resources to produce goods and services and to adapt to new market conditions caused by consumer demand or by innovation in the industry [31,37,44].

Adaptability has been studied as the process that activates innovation, to respond to market demands, so that studies of companies that are directed towards ecotourism, record evidence of

situations that require adaptation, and therefore stimulate the generation of a high degree of novelty in differential services, with strategies and processes that seek ways to add value through environmental and cultural wealth [18,19].

The capacity for innovation in tourism is one of the most effective tools in the development of this sector, emphasizing BigData projects. The tourism sector is one of the sectors that produces the most data on a daily basis and yet there are no major BigData studies to extract the most information from them. BigData's analysis in this sector is another of the innovative projects that this industry must implement and be studied by the academic community.

### 2.5. Relationship between Absorption and Innovation Dynamic Capabilities

The dynamic capacity of knowledge absorption includes processes that allows companies to "read" the market by analyzing information that, in turn, allows them to decode patterns and trends that impact the good or service they provide in their sector, as well as the reaction of their suppliers, competitors, related industries, customers and consumers in different regions or countries [5,39,48].

The challenge for companies is to develop the ability to accumulate, filter, leverage and apply market knowledge so that they can measure acceptance of their products and understand the impact of complementary activities that require collaborative work, both internally and externally, and can add value. This involves gathering and incorporating information that is available under specialized network knowledge structures, on a timely basis to allow for improved decision-making [7,31,37,43].

One of the ways to synthesize knowledge effectively is by developing the capacity for innovation, which has the virtue of establishing processes that take advantage of absorbed knowledge, to incorporate new functions or create new products, and also to mitigate the impact of negative externalities, identify new sources of raw materials, access new services that add value, enter new markets not served, improve operational processes or adopt new activities for commercialization and business management [16,17,41,48]. Studies of companies that identify the growth potential offered by ecotourism show that a capacity for knowledge absorption is required, which allows the interior of the company to manage knowledge with its employees, to read new consumption trends and take advantage of the market opportunities, to encourage creativity and development processes, which ultimately generate a capacity for innovation that manages to combine internal and external process, constituting a competitive advantage [16,19].

There is a lack of empirical studies to analyze the relationship of dynamic capabilities in contexts in which tourism is a relevant source of economic growth. The studies focused on this problem are of great importance, since there is a multitude of groups involved in the tourism sector, that is why the analyses must consider the heterogeneity of interest, needs, and impacts of different entrepreneurs at the SME level.

## 3. Problem Question and Hypotheses

With the purpose of evaluating the relationship of dynamic absorption, adaptation and innovation capacities at the SME level and at the cluster level that make up the tourism sector, the following problem question is proposed.

- What influence exists between the dynamic capabilities for the functioning of organizations in the tourism sector?

The previous question is supported by the following hypotheses:

**H1.** *There is a positive and significant relationship between dynamic absorption capability and dynamic adaptation capability.*

- **H1.1.** *There is a positive and significant relationship at the SME level between the dynamic absorption capacity and the dynamic adaptation capacity.*

- **H1.2.** *There is a positive and significant relationship at the level of the cluster between the dynamic absorption capability and the dynamic adaptation capability.*

**H2.** *There is a positive and significant relationship between dynamic adaptation capability and dynamic innovation capability.*

- **H2.1.** *There is a positive and significant relationship between dynamic adaptation capability and dynamic innovation capability at the individual SME level.*
- **H2.2.** *There is a positive and significant relationship between dynamic adaptation capability and dynamic innovation capability at the cluster level.*

**H3.** *There is a positive and significant relationship between dynamic absorption capability and dynamic innovation capability.*

- **H3.1.** *There is a positive and significant relationship between dynamic absorption capability and dynamic innovation capability at the individual SME level.*
- **H3.2.** *There is a positive and significant relationship between dynamic absorption capability and dynamic innovation capability at the cluster level.*

As shown in the following Figure 1, the proposed research model explains the possible interaction of dynamic capabilities, both at the level of individual SMEs and within clusters of these companies:

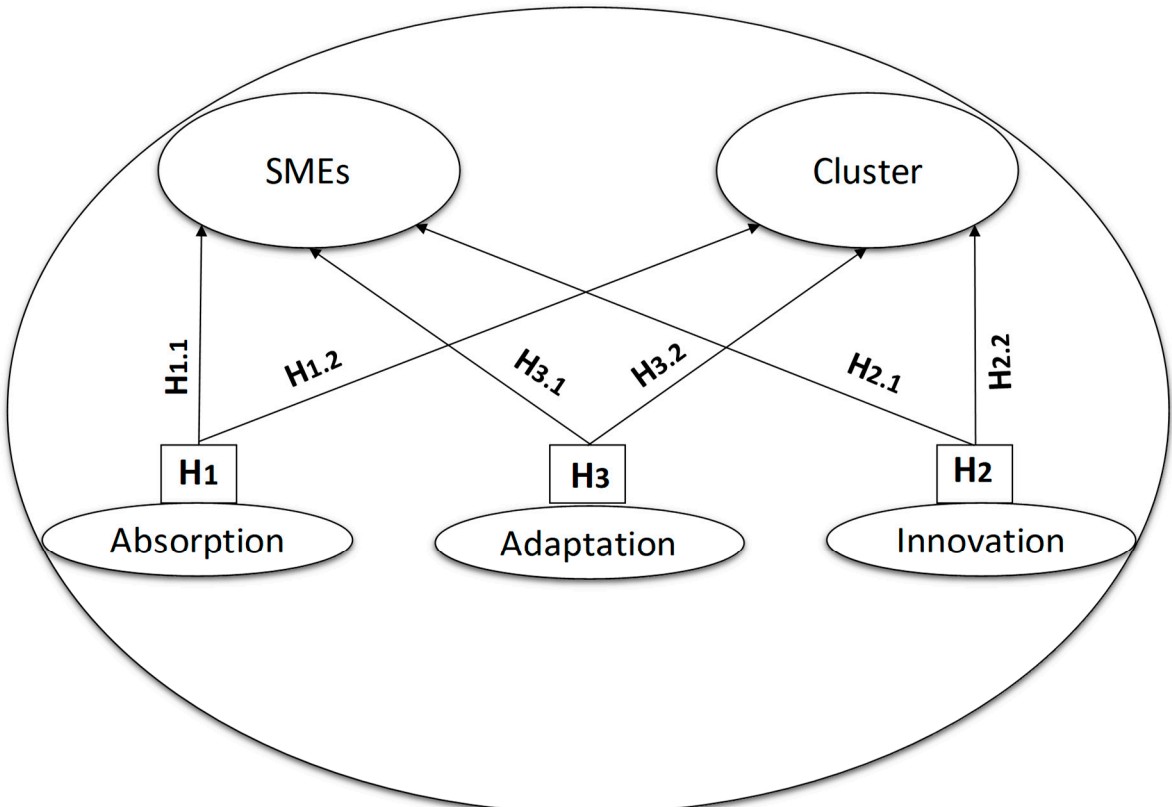

**Figure 1.** Research model (Source: created by the authors).

The previous model also reflects the evolution of contributions made by several authors, as well as the need to establish new approaches to the management and entrepreneurship of organizations (SMEs and Cluster in this document), which increase the importance of working under prisms of an integrative framework [49], and which David Teece (2019) addresses with more recent precision, under a new approach to capacity theory [50].

The combination of these dynamic capacities requires additional empirical evidence that the academic world tries to investigate [50,51], therefore this work is directed in the line proposed by the academic literature based on resource management, the role of managers and the organizational culture of innovation and entrepreneurship [52].

## 4. Method

### 4.1. Research Design, Sampling, and Data Collection

This cross-sectional study was carried out with small and medium enterprises attached to the nature tourism group of the Department of Tolima in Colombia, involving 230 entrepreneurs from 564 SMEs registered as of May 31, 2019 (106 hotels and rural areas tourist houses, 74 travel agencies, 17 tour guides and operators, 15 bars and restaurants, five tourist transport services companies and three resorts). SMEs were randomly selected from the directory of the National Tourism Registry of the Ministry of Commerce, Industry and Tourism [Ministerio de Comercio, Industria y Turismo], which was available as of the date of the study at http://www.rues.org.co/RNT. The sample size is sufficient to calculate a 95% confidence level and a 5% sample error, which are acceptable levels.

A structured survey instrument was developed in Spanish to measure the constructs in the research model. The questionnaire was provided via a personal invitation, addressed to executives and owners of SMEs based on the information in the national tourism registry who had worked during the last five years. The participants consisted of 54% men and 46% women, ranging from 31 to 72 years old.

### 4.2. Survey Instrument

Likert scales of five points ranging from 1 = not performed (never) to 5 = always performed (monthly), were developed based on previous research to measure each construct in the research model, taking different studies to describe and validate different dynamic capabilities. A pilot test was carried out on 25 managers or owners of SMEs belonging to the nature tourism group of Tolima (Colombia), located in the city of Ibague, which allowed us to discard or modify certain elements (we excluded elements that were not easily understood by the pilot survey participants and modified elements that were not appropriate at the cluster level), which allowed us to construct the final questionnaire that included six sections with 33 items, organized to measure each dynamic capacity at the SME level, as well as at the cluster level. Subsequently, the questionnaire was applied to 230 owners and businessmen, requesting that their answers be based on the current status of their businesses and the cluster.

The statistical program SPSS was used for data entry and basic results, while inferential analysis was performed using the AMOS 22 extension for modeling structural equations.

### 4.3. Relationship of Dynamic Capabilities at the SMEs Level

Wang and Ahmed (2007) measure the dynamic capacities of adaptation, absorption and innovation according to the following selection criteria:

- The scale should conform to the theoretical concept.
- The measurements have been used in other studies with samples similar to the present study.
- The scale facilitates the standardization of the questionnaire to facilitate tabulation and analysis of the data collected.

Based on Brikinshaw and Gibson (2004) the dynamic capacity of adaptation can be analyzed by first recording measurements at the company level, analyzing the difficulty of achieving flexibility. The authors studied 4,195 respondents in 41 business units of 10 multinational companies over three years, where they identified four behaviors, each of which involves taking independent and adaptive steps in the service of the general objectives of the company. They compared these measurements to companies such as Nokia, Ericsson, Oracle and Renault, which had performed strongly during that

period. This resulted in a proposal to managers regarding how to create these dynamic capacities in their own companies [53]. The study was cited in corporations such as SoftBank of Japan, a company that has achieved business innovation worldwide and has acquired knowledge about organizational adaptability that drives strategic innovation [54]. The study was also noted in work presented by a study done by NASA that reveals organizational dexterity as a process contingent on the route taken, instead of something attainable through generic applications of models of structural, temporal or ambidexterity models contextual [55].

Several studies contextualize absorption capacity as that condition in which a company recognizes the value of new and external information, assimilate it and apply it for commercial purposes, to generate innovative capabilities. Measuring absorption capacity is related to the degree of prior knowledge in the company, and to the market, beginning with an individual's cognitive base, which includes previous knowledge and background that are subsequently validated or contradicted by other individuals in the organization who constitute the diversity of an organization, which affects the investment in research and development that is, R&D [50,56,57].

The results of multiple works show how dynamic capacities influence the adoption and maintenance of environmental management and economic development processes, through the operationalization of internal and external capacities, ranging from simple to complex routines [26], generating the combination and reconfiguration of resources, through the absorption of knowledge, adaptation and innovation. The findings in the studies of ecotourism companies, also identify background, which allow discussing the difference between performance and the degree of impact that companies had on communities and the environment, for which the decisions taken in each case, encouraged the constitution of capacities that improved the operation and focus on sustainable development [13–16,18,19,27].

### 4.4. Relationships Among Dynamic Capabilities at the Cluster Level

At the level of networks and clusters, most scholars rely on the contributions of Brikinshaw and Gibson (2004) in attempting to explain the importance of dynamic adaptability and ambidexterity in complex organizations and supply chains. Their balanced theory of port competitiveness (BTOPC) relied on semi-structured interviews with professionals and experts from Pennsylvania State ports [58], as well as on supply chains and their contributions to organizational learning and resources of the association to adapt, using the read capacity and measuring its impact on operational performance, through an empirical analysis of the survey data carried out at high executives of manufacturing companies, which support the conventional wisdom that relates the collaboration of the supply chain with Lean and its contribution to improve operational performance [59].

The concept of dynamic organizational capacity for innovation has, in recent decades, been oriented toward sustainability (SOI), showing common patterns in companies that are collaborating. These patterns show three different levels of interconnected dynamic capabilities (adaptation, expansion and transformation) with respect to sustainability that explain a generative variation of innovative change and adaptation [25]. The dynamic capacity for adaptation is related to the exploitation and deployment of resources, supported by the acquisition, internalization and dissemination of knowledge, and the reconfiguration, divestment and integration of those resources. The dynamic capacity for innovation is related to the creation of completely new capabilities through exploration and route creation processes, backed by research, experimentation and risk taking, as well as the selection, financing and implementation of projects [60].

Studies on clusters affirm that a company's specific characteristics should be considered to explain innovation in clusters, since the combinations of internal and relational resources that make a strategy more effective can lead to a greater degree of innovation and partial modeling in the dynamics of clusters, which can access the absorption capacity of the companies in the cluster [61]. Other studies show that innovation capacity is related to improved economic quality in a region and also directly affects that region's international competitiveness. That competitiveness is achieved through effective

business management, including appropriate labor costs and R&D expenditures for scientific and technological efforts that require a high degree of specialized knowledge supplied by processes that are derived from the capacity for dynamic absorption [62].

*4.5. Analysis Data of the Measurement Model*

To analyze and explain the relationships between dynamic capabilities we use SPSS and AMOS for SEM, which allows us to specify, estimate, evaluate and validate the model shown in Figure 1, which demonstrates the hypothetical relationships between dynamic capabilities at the SME level and the tourism cluster. The degree of the relationship among the variables can be observed with their latent indicators or empirical variables (Appendix A).

We develop and validate the model that explores the relationship between observable variables and their measurement indicators to test whether the model fits well. We then load the data into the structural equation model to see the relationship among the endogenous variables. The modeling was done in two stages: first, we develop and validate the measurement model; then we analyze the data in the structural equation model using the surveys of 33 questions (15 questions for SMEs and 18 for the tourism cluster) related to dynamic absorption, adaptation and innovation capabilities.

Given that no question has a lower value than 0.300 in the standard regression weights, the model fit was confirmed using different adjustment indices, such as GFI (goodness of fit index), RMSR (residual mean square root) and means of quadratic error approximation (RMSEA).

## 5. Results and Discussion

*5.1. Relationship of Dynamic Capabilities in the Operation of SMEs*

The next Table 1 shows the correlation between the variables for SMEs, for each question in the questionnaire.

**Table 1.** Correlation matrix—SMEs level (Source: elaborated by the authors).

| Standardized Regression Weights | | | Estimate |
|---|---|---|---|
| P5.a | <— | Innovation | 0.499 |
| P5.b | <— | Innovation | 0.712 |
| P5.c | <— | Innovation | 0.72 |
| P5.d | <— | Innovation | 0.805 |
| P5.e | <— | Innovation | 0.853 |
| P6.a | <— | Adaptation | 0.856 |
| P6.b | <— | Adaptation | 0.769 |
| P6.c | <— | Adaptation | 0.774 |
| P6.d | <— | Adaptation | 0.824 |
| P6.e | <— | Adaptation | 0.886 |
| P7.a | <— | Absorption | 0.709 |
| P7.b | <— | Absorption | 0.763 |
| P7.c | <— | Absorption | 0.681 |
| P7.d | <— | Absorption | 0.799 |
| P7.e | <— | Absorption | 0.792 |

The variables observed (the answers to the questions) for the SMEs, which register greater relevance in the latent variables (dynamic capacities), are evidenced by the identification and rapid use of opportunities, r = 0.886 (P6.e) and the identification of sales opportunities of the current businesses, r = 0.856 (P6.a) with the dynamic adaptive capacity.

Likewise, SMEs show a strong relationship in the variables of formalization of alliances of suppliers and partners, r = 0.853 (P5.e) and in the acquisition of new technologies and methods, r = 0.824 (P5. D) with the dynamic innovation capacity.

It is also evident that SMEs have a medium-high relevance with respect to incentives for process implementation and process improvements, r = 0.799 (P7.d), and for dialogue with stakeholders to identify opportunities, r = 0.792, with dynamic absorption capacity.

Figure 2 shows the structural equation model that explains the possible interaction of the dynamic capabilities at the SME level.

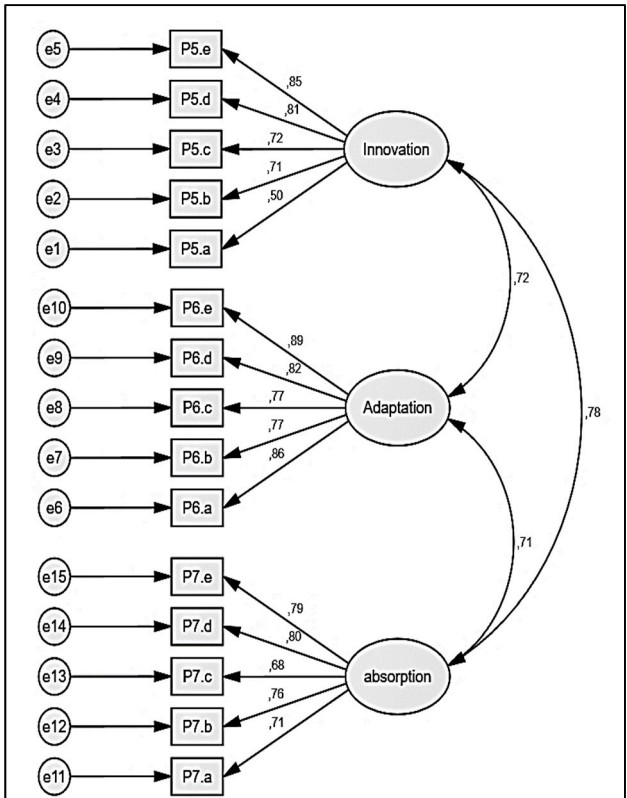

**Figure 2.** Model of standardized maximum likelihood estimates—SMEs. (Source: created by the authors with SPSS-AMOS 22).

There is a high correlation (r = 0.78) between the dynamic capacity of absorption and the dynamic capacity of innovation, while there is a medium-high relationship between the dynamic capacities of adaptation and innovation (r = 0.72), as well as between the dynamic capacity of absorption and the dynamic capacity of adaptation (r = 0.71).

For the SMEs, the strong relationship between innovation and knowledge absorption allows us to infer that the decisions of the owners and managers take into account an analysis of market and company information that allows them to innovate in their processes and tourist services. In the same way, we infer that SMEs must work to link innovations to the market or to the customer segment, as well as to improve their ability to take advantage of opportunities and to adapt personnel effectively throughout their processes to provide services more effectively.

### 5.2. List of Dynamic Capabilities in the Tourism Cluster

The following Table 2 shows the correlations between the observed variables and the endogenous variables of the cluster.

**Table 2.** Correlation matrix—cluster level (Source: created by the authors).

| Standardized Regression Weights | | | Estimate |
|---|---|---|---|
| P8.a | <—- | Absorption | 0.762 |
| P8.b | <— | Absorption | 0.735 |
| P8.c | <— | Absorption | 0.791 |
| P8.d | <— | Absorption | 0.855 |
| P8.e | <— | Absorption | 0.895 |
| P8.f | <— | Absorption | 0.849 |
| P9.a | <— | Adaptation | 0.735 |
| P9.b | <— | Adaptation | 0.803 |
| P9.c | <— | Adaptation | 0.807 |
| P9.d | <— | Adaptation | 0.802 |
| P9.e | <— | Adaptation | 0.836 |
| P9.f | <— | Adaptation | 0.866 |
| P10.a | <— | Innovation | 0.812 |
| P10.b | <— | Innovation | 0.841 |
| P10.c | <— | Innovation | 0.806 |
| P10.d | <— | Innovation | 0.893 |
| P10.e | <— | Innovation | 0.822 |
| P10.f | <— | Innovation | 0.653 |

The observed variables (questions) in the tourism cluster show a strong correlation between dynamic absorption capacity and decision-making regarding new markets and businesses based on information from interested parties, with an r = 0.895 (P8.e), and the use of technical methods to achieve effective commercial processes, with an r = 0.855 (P8.d).

On the other hand, tourism clusters show a high correlation with the dynamic capacity for innovation with respect to commercialization adjustment and business processes, with an r = 0.893 (P10.d), and the acquisition of new sources of raw materials, with an r = 0.822 (P10 e).

The tourism clusters also register an average relevance for shared and combined experiences among members in order to take advantage of opportunities, with an r = 0.866 (P9.f), and in the generation of new commercial activities with an r = 0.836 (P9.e), with respect to dynamic adaptive capacity.

Figure 3 shows the structural equation model that explains the possible interaction of the dynamic capabilities at the cluster level.

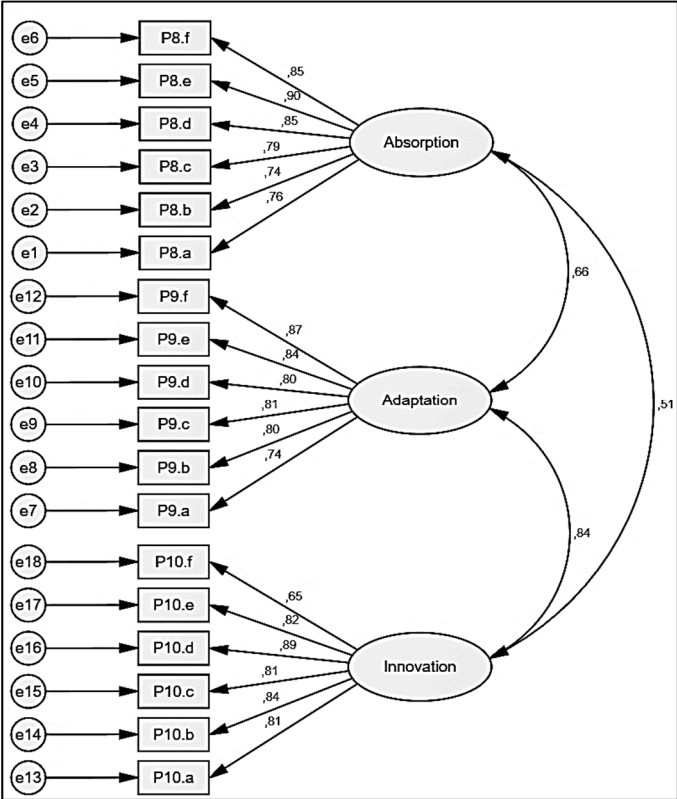

**Figure 3.** Model of standardized maximum likelihood estimates—cluster. (Source: created by the authors with SPSS-AMOS 22).

The endogenous variables in the previous model show a strong relationship between the dynamic capacity of adaptation and the dynamic capacity of innovation in the tourism cluster, with an r = 0.84, while there is a medium-low relationship between the dynamic capacities of absorption-adaptation and absorption-innovation, with an r = 0.66 and r = 0.51, respectively.

In the tourist cluster, the observed variables point to a high relationship between the dynamic capacities of adaptation and innovation, which allows us to infer, that in the decisions and actions undertaken by the cluster, there are processes of innovation that allow the cluster to adapt to new market rules once they are manifested. Since there is a medium-low relationship between dynamic absorption-innovation and absorption-adaptation capabilities, a lack of collaborative work may interfere with participation in local, national and international standards, and may prevent a cluster from generating high quality products, represented in the cluster by a culture of innovation based on imitating processes and services, as well as by alliances between interested parties.

*5.3. Discussion*

Many studies have analyzed the relationships between dynamic absorption, adaptation and innovation capabilities at different times, for example, Teece, et al. [3,22,23,28,35,46,47,50,52] that find a positive relationship between the three dynamic capacities in both individual companies and organizations, such as the tourism group. Our results are similar to previous studies, especially in the decisions taken by managers for the operation of organizations, whether SMEs or groups, as well as, is evidenced in the following Table 3.

**Table 3.** Summary results of the relationship between dynamic capabilities (Source: created by the authors).

| | **HYPOTHESES** | |
|---|---|---|
| **H.1** | **There is a positive and significant relationship between dynamic absorption capability and dynamic adaptation capability.** | |
| **H.1.1** | There is a positive and significant relationship at the SME level. ($\beta = 0.71$, $p < 0.01$). | Accept |
| **H.1.2** | There is a positive and significant relationship at the level of the cluster. ($\beta = 0.66$, $p < 0.01$). | Supported |
| **H.2** | **There is a positive and significant relationship between dynamic adaptation capability and dynamic innovation capability.** | |
| **H.2.1** | There is a positive and significant relationship at the individual SME level. ($\beta = 0.72$, $p < 0.01$). | Accept |
| **H.2.2** | There is a positive and significant relationship at the cluster level. ($\beta = 0.84$, $p < 0.01$). | Accept |
| **H.3** | **There is a positive and significant relationship between dynamic absorption capability and dynamic innovation capability.** | |
| **H.3.1** | There is a positive and significant relationship at the individual SME level. ($\beta = 0.78$, $p < 0.01$). | Accept |
| **H.3.2** | There is a positive and significant relationship at the cluster level. ($\beta = 0.51$, $p < 0.01$). | Supported |

Therefore, H1, H2 and H3 of the model shown are accepted, since we find a significant relationship between the dynamic capacities of absorption, adaptation and innovation, whose magnitude varies depending on whether the relationship is measured at the level of the SMEs or at the cluster level, and that allows validate the influence of the new sustainable tourism narratives in the decisions and actions undertaken, to adjust the functioning of the collaborative processes with employees, companies and stakeholders. The hypothesis validation model in our research, allows to ensure the need to link a new logic to develop business management models based on absorption, adaptation and innovation capabilities [3,22,23,28,35,46,47,50,52], and that are consistent with the relevance of the intake of decisions at the level of SMEs and complex groups such as clusters, highlighting the opportunity to use the model of dynamic capabilities, to facilitate the migration of traditional companies towards ecotourism, as a viable alternative to achieve sustainable local tourism [16–20].

In other words, an SME can move from being a traditional tourism company to an ecotourism company, by adopting a dynamic capabilities model [26], which facilitates decision-making around sustainable development, for which it requires appropriating a language that simplifies processes, develops routines and builds knowledge management (absorption), adaptation and innovation capabilities, capabilities that together constitute competitive advantages [24]. In addition, operational decisions to migrate to ecotourism under a dynamic capabilities model, allow improving the economic performance, in addition to mitigating the negative impacts on communities and local nature, which effectively allows the region to endure over time [16–18].

The presence of dynamic capability absorption in complex organizations [53] is necessary for adaptation within the logistics chains, since learning processes facilitate the orientation and allocation of resources, and improve the performance of value chains [50,59]. The seminal studies of Teece, et. to the. (1994, 1997), reflect the importance of structuring processes that direct the operation of the company, for which it is necessary to promote the dynamic capacity of innovation that is responsible for directing strategic sustainability efforts [25,35,46,54], and when combined with absorption and adaptation capabilities, ecotourism SMEs achieve superior economic returns, while generating benefits for the environment and stakeholders [16–20]. Our results, both in SMEs and in clusters, are similar in showing the positive relationship between dynamic capabilities.

The dynamic capacity model in ecotourism also requires collaborative work between SMEs, their owner-managers, their employees and interest groups, since the objectives cannot be achieved individually, and for this reason, it is that those interested, adopting dynamic capabilities model strengthen relationships from local groups [18,19,23].

On the other hand, cluster managers, according to the results obtained, must build new sustainable tourism narratives that promote the absorption of knowledge to innovate their products and operations,

guaranteeing a fair burden of tourists in the region, as well as to take advantage of the information provided and adapt quickly to the needs of the sustainable market, creating value for the communities and the ecosystem, based on the lessons learned in the innovation of tourism services of the SMEs.

## 6. Conclusions

The purpose of our study was to validate three dynamic capabilities in both SMEs and tourism clusters consisting of these companies, using observations of the dynamic capabilities absorption, adaptation and innovation. We collected data from tourism SMEs and the cluster they comprise, using a questionnaire as the primary data collection technique. Through these observed variables, our model allowed us to measure the relationship of generic dynamic capabilities, such as endogenous variables using SPSS-AMOS.

Bibliography on these dynamic capabilities had already pointed to factors in SMEs, but academic literature has not paid as much attention to analyzing the relationship of dynamic capabilities in contexts in which tourism is an obvious economic sector of wealth and development. This research has also confirmed the influence of absorption, adaptation and innovation capabilities in the sustainable development of the tourism sector.

The results clearly show that the endogenous variables have a significant direct relationship with dynamic absorption and innovation capacities in the tourism SMEs, which allows, through learning, to generate new processes and services, while the relationship between dynamic adaptation and innovation capabilities is more significant at the cluster level, impacting the operation of the cluster in its ability to adjust to changes in the market. Likewise, SMEs show a significant direct relationship to dynamic absorption-innovation and adaptation-innovation capabilities, reflecting that in decision-making for their operation, they seek to differentiate themselves with innovative services and processes.

Finally, the tourism cluster shows a medium relationship with respect to the dynamic capacities of absorption-adaptation and absorption-innovation, denoting the lack of collective and collaborative processes of learning and adaptation. We find that absorption, adaptation and innovation have positive, high and medium correlations at the level of the SMEs as the tourism cluster, requiring that owners and managers at each level develop new sustainability narratives, which allow optimizing the benefits that they provide the dynamic capacities of absorption, adaptation and innovation in the creation of competitive advantages in the different sustainable tourism services.

The current challenge lies in finding a way to make all of these capabilities interoperable with each other for maximum efficiency, this is precisely where we find the main obstacles. SMEs, or work with outdated systems or closed systems that do not allow interaction with new platforms and functionalities and the flow of information collected in one sector is not communicated to another sector that with that data could improve its operation, reinforcing the idea of promoting BigData projects in the tourism sector to extract the maximum information from them.

### 6.1. Limitations and Future Directions

There are limitations in this study according to the scope of the proposed project, although the three dynamic capacities discussed in this research had positive results, there are other capabilities that incorporate other factors, which can determine new levels of influence on the operation of companies, according to the context in which they operate.

Likewise, in the face of the multiple emerging trends of sustainable tourism, another of the current limitations is the short time that companies in the sector have been migrating from traditional tourism to new trends in ecotourism, so it is required to follow up later, to the decisions that entrepreneurs continue to adopt, to improve the dynamic capacity model, as well as new narratives of tourism sustainability.

Finally, the dynamic capabilities model was validated for Colombia in a region that was the epicenter of the armed conflict 60 years ago, the owners and generates of the SMEs and the Tolima nature tourism cluster, have the conviction to learn to adapt and innovating towards ecotourism, it is

clear that the richness of natural and cultural biodiversity is unique, which is why another limitation of our research is that it is impossible to affirm that the model is replicable to other regions, SMEs and Clusters, with other types of business.

In future studies, it is suggested that researchers validate whether the model is applicable to other regions and sectors committed to sustainable development, which may have clean production practices, circular economy and shared value creation. In addition, a complementary investigation is suggested in which it could use other latent variables, such as the dynamic capacities of internationalization, expansion, transformation, strategic sustainability, networks, among others, on the other hand, an investigation is suggested that evidences the impact of migrating from a traditional tourism to a sustainable ecotourism.

*6.2. Practical and Theoretical Contributions*

This study offers numerous practical and theoretical contributions. Specifically, it is useful for SMEs in the tourism sector, which can improve decision-making through the adoption of dynamic capabilities, such as those identified in this study, generating new sustainable narratives with the teams of collaborators. This study also provides guidance for cluster administrators, as it highlights the need to work collaboratively to improve the absorption of knowledge that allows the cluster to identify opportunities and combine and reconfigure processes and resources to adapt and innovate sustainable tourism services in the region. Similarly, such collaboration allows SMEs and cluster administrators to interact to ensure that employees practice and learn skills that improve performance, without affecting communities and the ecosystem. This suggests the importance of providing seminars and training workshops that foster the integration of dynamic capabilities in the generation of new sustainable narratives. Managers must also create a system to encourage the generation of new ideas that are implemented at both levels (SMEs and groups). SMEs must learn to work in networks to promote circles of trust, allowing groups to make joint decisions to innovate at times that need to adapt to the new demands of the sustainable consumption market, using underlying processes in the capacity for dynamic knowledge absorption, which would allow them to identify collective opportunities, exploit innovative marketing processes and services and generate sustainable benefits with different interest groups.

These empirical contributions highlight the importance of the evolution of the theoretical contributions of Teece, et. al., at the level of SMEs and clusters. The present study tries to reduce the gap between the conceptual definitions of dynamic capabilities with the empirical variables that allowed us to build our model, to achieve conclusive and decisive results for sectors that are growing worldwide, such as ecotourism. While there is room for additional confirmation, it is clear that dynamic capabilities absorption, adaptation and innovation, taken as endogenous latent variables, can be applied to both SMEs and clusters, without losing conceptual rigor.

**Author Contributions:** A.J.G.R. initiated the paper; A.J.G.R., J.M.G.M. they compiled the theoretical framework, A.J.G.R., N.J.B. and J.M.G.M. jointly developed the methodology; A.J.G.R. Design of the model in Figure 1 and hypothesis, N.J.B. Development of the structural equation model, J.M.G.M. validated the model, A.J.G.R. and N.J.B. joint work of data processing and analysis of results; A.J.G.R. instrument design and questionnaire application, N.J.B. Tabulation, application of the model of structural equations, A.J.G.R., N.J.B. and J.M.G.M. overall editing of the paper, This being a collective effort. All authors read and approved the final manuscript.

**Funding:** This research was funded by Universidad de Ibague.

**Acknowledgments:** We thank the executives and owners of the SMEs, which make up the Tourism Cluster of the Tolima, who participated anonymously, for their generous support for this study. This work was supported by the Universidad de Ibague, in Colombia, within the framework of the research project strategic update of the Tourism Cluster of Tolima.

**Conflicts of Interest:** The authors declare no conflict of interest.

## Appendix A

| | P5.a | P5.b | P5.c | P5.d | P5.e | P6.a | P6.b | P6.c | P6.d | P6.e | P7.a | P7.b | P7.c | P7.d | P7.e | P8.a | P8.b | P8.c | P8.d | P8.e | P8.f | P9.a | P9.b | P9.c | P9.d | P9.e | P9.f | P10.a | P10.b | P10.c | P10.d | P10.e | P10.f |
|---|---|---|---|---|---|---|---|---|---|---|---|---|---|---|---|---|---|---|---|---|---|---|---|---|---|---|---|---|---|---|---|---|---|
| P5.a | 1 | | | | | | | | | | | | | | | | | | | | | | | | | | | | | | | | |
| P5.b | 0,756 | 1 | | | | | | | | | | | | | | | | | | | | | | | | | | | | | | | |
| P5.c | 0,795 | 0,799 | 1 | | | | | | | | | | | | | | | | | | | | | | | | | | | | | | |
| P5.d | 0,812 | 0,816 | 0,857 | 1 | | | | | | | | | | | | | | | | | | | | | | | | | | | | | |
| P5.e | 0 | 0 | 0 | 0 | 1 | | | | | | | | | | | | | | | | | | | | | | | | | | | | |
| P6.a | 0 | 0 | 0 | 0 | 0,357 | 1 | | | | | | | | | | | | | | | | | | | | | | | | | | | |
| P6.b | 0 | 0 | 0 | 0 | 0,395 | 0,399 | 1 | | | | | | | | | | | | | | | | | | | | | | | | | | |
| P6.c | 0 | 0 | 0 | 0 | 0,447 | 0,451 | 0,498 | 1 | | | | | | | | | | | | | | | | | | | | | | | | | |
| P6.d | 0 | 0 | 0 | 0 | 0 | 0 | 0 | 0 | 1 | | | | | | | | | | | | | | | | | | | | | | | | |
| P6.e | 0 | 0 | 0 | 0 | 0 | 0 | 0 | 0 | 0,516 | 1 | | | | | | | | | | | | | | | | | | | | | | | |
| P7.a | 0 | 0 | 0 | 0 | 0 | 0 | 0 | 0 | 0,6 | 0,538 | 1 | | | | | | | | | | | | | | | | | | | | | | |
| P7.b | 0 | 0 | 0 | 0 | 0 | 0 | 0 | 0 | 0,595 | 0,534 | 0,62 | 1 | | | | | | | | | | | | | | | | | | | | | |
| P7.c | -0,606 | -0,61 | -0,642 | -0,658 | -0,144 | -0,146 | -0,162 | -0,185 | 0 | 0 | 0 | 0 | 1 | | | | | | | | | | | | | | | | | | | | |
| P7.d | -0,59 | -0,593 | -0,624 | -0,64 | 0 | 0 | 0 | 0 | -0,3 | -0,268 | -0,314 | -0,311 | 0,793 | 1 | | | | | | | | | | | | | | | | | | | |
| P7.e | 0 | 0 | 0 | 0 | -0,174 | -0,176 | -0,195 | -0,223 | -0,372 | -0,332 | -0,389 | -0,386 | 0,318 | 0,583 | 1 | | | | | | | | | | | | | | | | | | |
| P8.a | -0,611 | -0,615 | -0,657 | -0,677 | 0 | 0 | 0 | 0 | 0 | 0 | 0 | 0 | 0,942 | 0,819 | 0,184 | 1 | | | | | | | | | | | | | | | | | |
| P8.b | 0 | 0 | 0 | 0 | -0,401 | -0,405 | -0,449 | -0,511 | 0 | 0 | 0 | 0 | 0,556 | 0,25 | 0,673 | 0,137 | 1 | | | | | | | | | | | | | | | | |
| P8.c | 0 | 0 | 0 | 0 | 0 | 0 | 0 | 0 | -0,636 | -0,569 | -0,664 | -0,658 | 0,228 | 0,612 | 0,759 | 0,122 | 0,205 | 1 | | | | | | | | | | | | | | | |
| P8.d | 0,081 | 0,081 | 0,086 | 0,089 | 0 | 0 | 0 | 0 | 0 | 0 | 0 | 0 | -0,062 | -0,061 | 0 | -0,085 | 0 | 0 | 1 | | | | | | | | | | | | | | |
| P8.e | -0,088 | 0,001 | 0,003 | 0,006 | 0 | 0 | 0 | 0 | 0 | 0 | 0 | 0 | 0,008 | 0,008 | 0 | 0,004 | 0 | 0 | -0,004 | 1 | | | | | | | | | | | | | |
| P8.f | 0,001 | -0,089 | 0,003 | 0,007 | 0 | 0 | 0 | 0 | 0 | 0 | 0 | 0 | 0,009 | 0,006 | 0 | 0,005 | 0 | 0 | -0,004 | -0,014 | 1 | | | | | | | | | | | | |
| P9.a | 0,002 | 0,002 | -0,106 | 0,013 | 0 | 0 | 0 | 0 | 0 | 0 | 0 | 0 | 0,017 | 0,017 | 0 | 0,009 | 0 | 0 | -0,008 | -0,025 | -0,03 | 1 | | | | | | | | | | | |
| P9.b | 0,003 | 0,004 | 0,011 | -0,125 | 0 | 0 | 0 | 0 | 0 | 0 | 0 | 0 | 0,033 | 0,032 | 0 | 0,018 | 0 | 0 | -0,016 | -0,054 | -0,087 | -0,114 | 1 | | | | | | | | | | |
| P9.c | 0 | 0 | 0 | 0 | 0,137 | 0,139 | 0,156 | 0,188 | 0 | 0 | 0 | 0 | -0,045 | 0 | -0,054 | 0 | -0,139 | 0 | 0 | 0 | 0 | 0 | 0 | 1 | | | | | | | | | |
| P9.d | 0 | 0 | 0 | 0 | -0,121 | -0,002 | -0,001 | 0,004 | 0 | 0 | 0 | 0 | 0,006 | 0 | 0,007 | 0 | 0,01 | 0 | 0 | 0 | 0 | 0 | 0 | -0,029 | 1 | | | | | | | | |
| P9.e | 0 | 0 | 0 | 0 | -0,002 | -0,123 | -0,002 | 0,004 | 0 | 0 | 0 | 0 | 0,006 | 0 | 0,008 | 0 | 0,01 | 0 | 0 | 0 | 0 | 0 | 0 | -0,03 | -0,013 | 1 | | | | | | | |
| P9.f | 0 | 0 | 0 | 0 | -0,004 | -0,004 | -0,147 | 0,006 | 0 | 0 | 0 | 0 | 0,01 | 0 | 0,012 | 0 | 0,016 | 0 | 0 | 0 | 0 | 0 | 0 | -0,046 | -0,02 | -0,021 | 1 | | | | | | |
| P10.a | 0 | 0 | 0 | 0 | -0,009 | -0,008 | -0,005 | -0,204 | 0 | 0 | 0 | 0 | 0,022 | 0 | 0,027 | 0 | 0,037 | 0 | 0 | 0 | 0 | 0 | 0 | -0,106 | -0,047 | -0,048 | -0,074 | 1 | | | | | |
| P10.b | 0 | 0 | 0 | 0 | 0 | 0 | 0 | 0 | 0,141 | 0,125 | 0,149 | 0,146 | 0 | -0,063 | -0,078 | 0 | 0 | -0,145 | 0 | 0 | 0 | 0 | 0 | 0 | 0 | 0 | 0 | 0 | 1 | | | | |
| P10.c | 0 | 0 | 0 | 0 | 0 | 0 | 0 | 0 | -0,149 | -0,001 | 0,004 | 0,004 | 0 | 0,015 | 0,019 | 0 | 0 | 0,015 | 0 | 0 | 0 | 0 | 0 | 0 | 0 | 0 | 0 | 0 | -0,023 | 1 | | | |
| P10.d | 0 | 0 | 0 | 0 | 0 | 0 | 0 | 0 | 0,001 | -0,123 | 0,002 | 0,002 | 0 | 0,006 | 0,011 | 0 | 0 | 0,008 | 0 | 0 | 0 | 0 | 0 | 0 | 0 | 0 | 0 | 0 | -0,013 | -0,019 | 1 | | |
| P10.e | 0 | 0 | 0 | 0 | 0 | 0 | 0 | 0 | 0,003 | -0,001 | -0,166 | 0,005 | 0 | 0,021 | 0,026 | 0 | 0 | 0,02 | 0 | 0 | 0 | 0 | 0 | 0 | 0 | 0 | 0 | 0 | -0,032 | -0,049 | -0,027 | 1 | |
| P10.f | 0 | 0 | 0 | 0 | 0 | 0 | 0 | 0 | 0,002 | -0,001 | 0,006 | -0,162 | 0 | 0,019 | 0,024 | 0 | 0 | 0,019 | 0 | 0 | 0 | 0 | 0 | 0 | 0 | 0 | 0 | 0 | -0,03 | -0,044 | -0,025 | -0,061 | 1 |

**Figure A1.** Correlations of Estimates (Default model).

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
