# Peer review of "Validity of Dynamic Capabilities in the Operation Based on New Sustainability Narratives on Nature Tourism SMEs and Clusters"

_sustainability, doi:10.3390/su12031004_

Round 1

Reviewer 1 Report

This reviewer feels that this manuscript was written well.   However, one major concern is that this reviewer was hard to find any justifcation of how the findings and implications of this study can contribute to sustainablity. For helping the authors, I would like to suggest several minor issues for its improvement of this study.

For the abstract, please be specific - three dynamic capacities.  What are the three dynamic capacities? For the abstract, " ... which 19 highlight the greatest contribution, which is to contribute an empirical study supported ..."  I think the sentence a bit unclear. For the abstract, "Our results also point out the limitations and challenges for the aggregate tourism sector in 25 Colombia."  The sentence is not consistent with previous senteces. While previous sentences focused on "dynamic capacities,"  the very last sentence newly introduced a new concept (i.e., the aggregate tourism sector). For introduction, the rationale/justification of this study looks a bit not so strong. Please clearly highlight research gaps and significance of this studF For "5.1. Limitations and future directions" and "5.2. Practical and theoretical contributions,"  I feel that "5.1. Practical and theoretical contributions" and "5.2. Limitations and future directions." Discussion part would be recommended to more expand than the current version. As I noted before, little information was discussed regarding sustainability.   This journal focuses on sustainability aspects.  

Author Response

Point 1: For the abstract, please be specific - three dynamic capacities.  What are the three dynamic capacities?

Response 1: Accepted point, and the three dynamic capabilities of the investigation were included in the abstract

Point 2: For the abstract, " ... which 19 highlight the greatest contribution, which is to contribute an empirical study supported ..."  I think the sentence a bit unclear. For the abstract, "Our results also point out the limitations and challenges for the aggregate tourism sector in 25 Colombia."  The sentence is not consistent with previous senteces. While previous sentences focused on "dynamic capacities,"  the very last sentence newly introduced a new concept (i.e., the aggregate tourism sector). 

Response 2: The points were accepted, and in the abstract the contribution of the manuscript was corrected, and the concept of tourism that we want to highlight was clarified

Point 3: For introduction, the rationale/justification of this study looks a bit not so strong. Please clearly highlight research gaps and significance of this studF For "5.1. Limitations and future directions" and "5.2. Practical and theoretical contributions,"  I feel that "5.1. Practical and theoretical contributions" and "5.2. Limitations and future directions." 

Response 3: The points were accepted, and we improve and clarify the points requested in the introduction; numbering and arguments of limitations, future addresses, contributions were adjusted

Point 4: Discussion part would be recommended to more expand than the current version. As I noted before, little information was discussed regarding sustainability.   This journal focuses on sustainability aspects. 

Response 4: The points were accepted, and we corrected the wording of the discussion, in aspects of tourism sustainability, from the concept of "ecotourism", which is the topic discussed within the questionnaires applied to owners and managers of SMEs

Reviewer 2 Report

This paper deals with the "validity of dynamic capabilities in the operation based on new sustainability narratives on Nature Tourism SMEs and Clusters." The authors tried to address how a specific group of businesses can promote Nature Tourism (aka. ecotourism). To improve the quality of this paper, I suggest the following points below. 

1. The Introduction has a weak rationale.

The authors strongly supported the significance of "travel competitiveness" and "dynamic capabilities of companies".   However, they seemed to be negligent to mention the significance of "ecotourism" or "Nature Tourism".Please ask "why ecotourism is important?" This part needs improvement in terms of delivery. This means the authors should try to explain the key rationales of this study to the wider audience. Please improve the readability of this article.

2. Literature Review and Hypothesis

I suggest authors should separate the Review and the Hypothesis in different sections (e.g. 2. Lit. Review + 3. Hypothesis). On top of that, in a new Hypothesis section, please add the research questions underpinning this research. (3. Hypothesis -> Research Questions and Hypothesis) The reviews only contain a summary of the empirical studies. Probably, the readers would want the authors' perceptive interpretations of these studies. Please add your assessment and reflections of these studies. 

3. Method

Figure 1 Research Model should be moved to 3.5 Analysis data of the measurement model. In 3.1, please add a table indicating the overview of the data profile. 

4. Results and Discussion

Elaborate the discussion from line #385-396. (sounds simplistic for the readers) on line #384, whose work? "the recent works of ___________ et al." (author missing)

Extra. 

-This article seems quite difficult to read. So, when you revise, please try to improve the clarity of the text. 

-Some minor grammatical errors spotted (such as article missing, spelling, and misuse of prepositions). Please work this out as much as you can.

Author Response

Point 1: 1. The Introduction has a weak rationale.

The authors strongly supported the significance of "travel competitiveness" and "dynamic capabilities of companies".   However, they seemed to be negligent to mention the significance of "ecotourism" or "Nature Tourism".Please ask "why ecotourism is important?" This part needs improvement in terms of delivery. This means the authors should try to explain the key rationales of this study to the wider audience. Please improve the readability of this article.

Response 1: The observation is fully accepted, and we correct the introduction

Point 2: 2. Literature Review and Hypothesis

I suggest authors should separate the Review and the Hypothesis in different sections (e.g. 2. Lit. Review + 3. Hypothesis). On top of that, in a new Hypothesis section, please add the research questions underpinning this research. (3. Hypothesis -> Research Questions and Hypothesis) The reviews only contain a summary of the empirical studies. Probably, the readers would want the authors' perceptive interpretations of these studies. Please add your assessment and reflections of these studies. . 

Response 2: The observation is accepted, and we correct it by separating literature review and hypothesis. We also improve the writing and highlight the importance of the empirical work of our research

Point 3: 3. Method

Figure 1 Research Model should be moved to 3.5 Analysis data of the measurement model. In 3.1, please add a table indicating the overview of the data profile.  

Response 3: We accept the point, and move the figure towards the data analysis, and include table 3 with data profile obtained with SPSS Amos 22

Point 4: 4. Resultados y discusión

Elabore la discusión de la línea # 385-396. (suena simplista para los lectores) en la línea # 384, ¿de quién es el trabajo? "Los trabajos recientes de ___________ et al." (autor desaparecido)

Response 4: We accepted the point, and the discussion was improved and the error in line # 384 was corrected

Point extra: Extra. 

-This article seems quite difficult to read. So, when you revise, please try to improve the clarity of the text. 

-Some minor grammatical errors spotted (such as article missing, spelling, and misuse of prepositions). Please work this out as much as you can.  

Response 3: We accept the observations, and we review the errors of writing and grammar, hoping that they conform to the expected. Thank you

Round 2

Reviewer 2 Report

Thank you for addressing the comments I provided. 

Before submitting the final version, please address my comments below. 

2. Literature Review

-in each sub-section (2.1~2.5), you should add two sentences of assessment and reflection of the pieces of literature you considered. 

4. methods

-(pg 7) 3.2 Survey Instrument --> 4.2 Survey Instrument?

5.3 Discussion 

-(pg.13) Table 3 --> HYPÓTHESES (X) --> HYPOTHESES (O)

I think Spanish is not appropriate in this table. 

6. Conclusion

This section needs improvement.

Especially, 6.2 and 6.3 are not sufficient for having a sub-section heading. 

(insufficient amount of content to have these sub-sections)

--> In other words, the authors did neither sufficiently explain the limitation, not sufficiently suggest the future direction.

My suggestion is to take all the sub-section headings off (6.1 to 6.3). Also, please revise the arguments in the conclusion section to improve the quality of the presentation.  

Extra. spell and citation style checks are required. 

Author Response

Dear Reviewer:

We appreciate your valuable recommendations, which were fully accepted, and please see the attachment.
